# Clinical Characterization and Outcomes of Patients with Hypercreatinemia Affected by COVID-19

**DOI:** 10.3390/healthcare11070944

**Published:** 2023-03-24

**Authors:** Ahmed M. E. Elkhalifa, Naveed Nazir Shah, Zaid Khan, Sofi Imtiyaz Ali, Showkat Ul Nabi, Showkeen Muzamil Bashir, Masood Saleem Mir, Elsharif. A. Bazie, Abozer Y. Elderdery, Awadh Alanazi, Fawaz O. Alenazy, Elsadig Mohamed Ahmed

**Affiliations:** 1Department of Public Health, College of Health Sciences, Saudi Electronic University, Riyadh 11673, Saudi Arabia; 2Department of Haematology, Faculty of Medical Laboratory Sciences, University of El Imam El Mahdi, Kosti 1158, Sudan; 3Department of Chest Medicine, Government Medical College, Srinagar 191202, Jammu & Kashmir, India; 4Biochemistry & Molecular Biology Lab, Division of Veterinary Biochemistry, Faculty of Veterinary Sciences (F.V.Sc.) and Animal Husbandry (A.H), SKUAST-K, Shuhama, Alusteng, Srinagar 190006, Jammu & Kashmir, India; 5Large Animal Diagnostic Laboratory, Department of Clinical Veterinary Medicine, Ethics & Jurisprudence, Faculty of Veterinary Sciences (F.V.Sc.) and Animal Husbandry (A.H), SKUAST-K, Shuhama, Alusteng, Srinagar 190006, Jammu & Kashmir, India; 6Department of Veterinary Pathology, Faculty of Veterinary Sciences (F.V.Sc.) and Animal Husbandry (A.H), SKUAST-K, Shuhama, Alusteng, Srinagar 190006, Jammu & Kashmir, India; 7Pediatric Department, Faculty of Medicine University of El Imam El Mahdi, Kosti 1158, Sudan; 8Department of Clinical Laboratory Sciences, College of Applied Medical Sciences, Jouf University, Sakaka 72388, Saudi Arabia; 9Department of Medical Laboratory Sciences, College of Applied Medical Sciences, University of Bisha, P.O. Box 551, Bisha 61922, Saudi Arabia; 10Department of Clinical Chemistry, Faculty of Medical Laboratory Sciences, University of El Imam El Mahdi, Kosti 1158, Sudan

**Keywords:** COVID-19, hypercreatinemic, mechanical ventilation, intensive care, outcome

## Abstract

The present study evaluated the clinical presentation and outcome of COVID-19 patients with underlying hypercreatinemia at the time of hospitalization. A retrospective observational study was conducted from the 23rd of March 2020 to the 15th of April 2021 in 1668 patients confirmed positive for COVID-19 in the Chest Disease Hospital in Srinagar, India. The results of the present study revealed that out of 1668 patients, 339 with hypercreatinemia had significantly higher rates of admission to the intensive care unit (ICU), severe manifestations of the disease, need for mechanical ventilation, and all-cause mortality. Multivariable analysis revealed that age, elevated creatinine concentrations, IL-1, D-Dimer, and Hs-Crp were independent risk factors for in-hospital mortality. After adjusted analysis, the association of creatinine levels remained strongly predictive of all-cause, in-hospital mortality (HR-5.34; CI-4.89–8.17; *p* ≤ 0.001). The amelioration of kidney function may be an effective method for achieving creatinemic targets and, henceforth, might be beneficial for improving outcomes in patients with COVID-19.

## 1. Introduction

A pneumonic disease outbreak caused by a novel beta-coronavirus, named “severe acute respiratory syndrome coronavirus 2 (SARS-CoV-2), occurred in Wuhan, China [1,2,3]. The disease caused by the virus is known as the Corona Virus Infectious Disease 2019 (COVID-19) and represents a source of concern for the treatment of patients with associated co-morbidities. The clinical disease affects the respiratory system and is characterized by acute respiratory distress syndrome. In addition to pulmonary involvement, SARS-CoV-2 has been isolated from cardiovascular, digestive, neurological and renal tissues [4,5,6]. The incidence of renal failure varies from 0.5% to 25% in hospitalized patients in the absence of medical emergencies related to a pandemic [7,8]. According to the Kidney Disease Improving Global Outcome [9] guidelines, kidney dysfunction (KD) is characterized by known histological damage and a suboptimal glomerular filtration rate (GFR) lasting for at least three months. The center for Disease Control and Prevention estimated that 15% of adults have some degree of kidney dysfunction [9]. A study from China indicated that 4.3% of KD patients wereinfected with SARS-CoV-2 and that these patients manifested severe COVID-19 clinical presentation [10]. End-stage KD patients are a particularly vulnerable category with an infection rate of 16% of the population [11]. Patients with KD are more likely to suffer from other coexisting comorbidities, such as high blood pressure, diabetes, and CVDs. Hence, it is expected that patients with KD, particularly those on dialysis, will be at extreme risk during the ongoing COVID-19 pandemic [12]. As the pandemicprogresses, increased importance must be given to understanding disease presentation in high-risk subgroups for better patient management. Kidney Dysfunction (KD) is commonly found in the general population and these patients are at risk of increased viral morbidity and grave outcome. Some studies have reported increased mortality in COVID-19 patients hospitalized with associated kidney failure [13,14]. Furthermore, there are conflicting reports on the impact of kidney failure on the disease outcome of COVID-19, which might be due to variability in genetics and environmental factors of the different populations studied [1,15,16]. Severe manifestations of COVID-19 in patients with pre-existing renal dysfunction have been attributed to direct renal damage by SARS-CoV-2 [17,18] and secondary renal damage caused by thrombotic microangiopathy, hypo perfusion and sepsis [19]. Furthermore, some studies have identified patients with preexisting renal failure as having severe outcomes of COVID-19 [20]. Although the details concerning acute kidney injury caused by COVID-19 are comparatively comprehensive, few studies have explored the characteristics of COVID-19 in patients with pre-existing renal insufficiency. To date in India, we do not have any study or data on characteristics and outcomes of kidney disease associated with the COVID-19 pandemic. Our objective in the present study was to explore presentations of COVID-19 in patients with renal dysfunction so that possible medical attention and strategies can be evolved for early amelioration. Therefore, in the present study, we intended to examine theclinical, laboratory and disease outcome of hypercreatinemic COVID-19 patients.

## 2. Material and Method

### 2.1. Patients and Study Design

We carried out consecutive systematic sampling for one year (March 2020 to the 15 April 2021) in all the cases of COVID-19-positive patients in the Chest Disease Hospital in Srinagar, India. Confirmed COVID-19 patients were diagnosed on the clinical and radiological criteria established by the World Health Organization (WHO) and confirmed by RT-PCR for SARS-CoV-2 in respiratory samples [21]. Patients with histories of other co morbidities, other infections and patients in critical stages of a COVID-19 infection at the time of hospitalization were excluded from the study. The analysis was conducted in all cases with outcomes and a hospitalization of more than 4 weeks. We considered the baseline serum Cr value as the serum Cr level that was measured in the healthy population and, henceforth, standardized in our laboratory (Chest Disease Laboratory, Srinagar Jammu and Kashmir). KD (kidney dysfunction) diagnosis was based on Kidney disease guidelines as defined by Improving Global Outcomes (KDIGO) CKD Work Group (2012) classification [9]. KD was defined as per serum creatinine criteria and was defined as a significant increase in serum creatinine by 1.5 times higher from the baseline values established in Chest Disease Laboratory, Srinagar Jammu and Kashmir. The values were considered normal when they fell between reference values indicated in hospital laboratory. Patients were categorized on basis of creatinine levels as normocreatinemic and hypercreatinemic as per KDIGO, 2012 guidelines. The present investigation was conducted in accordance with the principles outlined in the Declaration of Helsinki and informed consent was obtained from the patients for use of datasheet and for the analysis of their clinical data, being reflected in the electronic datasheet. In order to differentiate patients which developed kidney dysfunction during hospitalization from those patients which were admitted with pre-existing kidney dysfunction, we excluded patients which developed kidney dysfunction during hospitalization.

### 2.2. Data Analysis

Comparison for continuous variables was estimated by Mann-Whitney U test and data were presented as mean with standard deviation, while categorical data were compared by χ^2^ test and presented as a percentage. Cox proportional hazard regression analysis was used to identify the independent associations between Cr levels and in-hospital mortality, disease severity and need for ICU admission in patients with COVID-19. The Kaplan-Meier method was used to assess cumulative rates of survival and to calculate statistical significance. A *p* value of less than 0.05 was considered statistically significant and statistics was performed with IBM SPSS Statistics, version 20.2.

## 3. Results

### 3.1. Baseline Characteristics of Patients on Admission and Outcome

A total of 1668 COVID-19 patients were included in the current investigation, with a mean age of 61.56 ± 21.62 years, among them 967 (57.97%) were males. Among these 167 (10.01%) had diabetes mellitus, 145 (8.69%) had hypertension, 21 (1.25%) had end-stage renal failure, 101 (6.05%) has cardiovascular dysfunctions, and 123 (7.37%) were having more than one co-morbidities prior to hospitalization. After the exclusion of these patients remaining 1,111 patients were categorized on basis of creatine levels at the time of hospitalization, 772 (69.48%) were hypercreatinemic (1.5 mg/dL) and 339 (30.51%) were having creatinine levels below the threshold level (Figure 1). The mean e-GFR of these patients was 98.56 ± 24.22 mL/min/1.73 m^2^; the mean eGFR in the hypercreatinemic-sub-population was 84.61 ± 19.12 mL/min/1.73 m^2^ while as in normocreatinemicsub-population eGFR was 96.90 ± 23.34 mL/min/1.73 m^2^. Similarly, in the hypercreatinemic sub-population mean creatinine levels were 1.87 ± 0.32 mg/dL, and in normocreatinemic sub-population levels of creatinine was 0.91 ± 0.13 mg/dL. The severity of COVID-19 disease was defined as per the guidelines of the World Health Organization classification for COVID-19 and the CURB-65 scale [21,22]. From Table 1 it can be observed that compared to the normocreatinemic sub-population, hypercreatinemic patients were of an older age group (67.43 ± 4.36: IQR-17 v/s 48.00 ± 4.05: IQR-20, *p* ≤ 0.05) and were having a higher number of males 189 (55.75%). Clinically, body temperature was found to be significantly (*p* ≤ 0.05) elevated in patients with hypercreatinemia (38.24 ± 0.40) compared to the normocreatinemic sub-population (36.76 ± 0.39). Furthermore, oxygen saturation (SPO_2_) was significantly reduced in hypercreatinemic patients compared to the normocreatinemic sub-population (83.56 ± 5.23 vs. 89.87 ± 3.14 *p* ≤ 0.01) (Table 1; Figure 1). The symptomatic profile of patients with hypercreatinemia was quite different compared to the normocreatinemic sub-population. The major symptoms of concern included productive sputum, rhinitis, diarrhea, headacheand fever, which were significantly higher in hypercreatinemic patients compared to the normocreatinemic sub-population (Table 2).

Biochemical parameters (Table 1) which include WBC count (6277.12 ± 871.23: v/s 8234.16 ± 512.34, *p* ≤ 0.05) and lymphocyte count (1456.1 ± 312.1 v/s 1522.3 ± 213.3, *p* ≤ 0.05) were significantly reduced in hypercreatinemic patients compared to the normocreatinemic sub-population which reflects some degree of immunosuppression in hypercreatinemic patients, hence their increased risk of contracting disease and severe manifestation of disease. Furthermore, at the time of hospitalization, important findings of the present study were significantly elevated levels of LDH, SGOT (IU/L) SGPT (IU/L) ALP (IU/L), while significantly reduced levels of proteins (8.24 ± 0.76; IQR-4.5 v/s 6.614 ± 0.32; IQR-1.9; *p* ≤ 0.05) were observed in hypercreatinemic patients compared to normocreatinemic patients. The acute phase response/pro-inflammatory response revealed Ferritin (*p* ≤ 0.05) (Figure 2), procalcitonin (*p* ≤ 0.05), IL-1 (*p* ≤ 0.01) (Figure 3) Hs-Crp (*p* ≤ 0.01) (Figure 4), and IL-2 (*p* ≤ 0.01) (Figure 5) were significantly higher in patients with hypercreatinemia compared to normocreatinemic patients. Furthermore, Ferritin (r^2^-0.065; *p* ≤ 0.05), Hs-Crp (r^2^-0.256; *p* ≤ 0.001), procalcitonin (r^2^-0.062; *p* ≤ 0.05), LDH (r^2^-0.195; *p* ≤ 0.01). IL-1 (r^2^-0.350; *p* ≤ 0.001) and IL-6 (r^2^-0.256; *p* ≤ 0.001) levels were correlated with creatinine levels at the time of hospitalization (Figure 6). The correlation studies indicate that elevated levels of creatinine were positively correlated with levels of pro-inflammatory cytokines which include Hs-Crp followed by IL-6, LDH and procalcitonin. These correlations studies indicate the role of creatinine in the activation of the immune cum inflammatory pathway which culminates into a cytokine storm. Similarly, it can be proposed that pro-inflammatory cytokines cause damage to renal tissue which leads to elevated levels of creatinine.

### 3.2. Outcome

The median time of hospitalization in patients with normal creatinine levels was 20.33 ± 3.56 days (IQR-12–27) while in patients with hypercreatinemia, median time of hospitalization was 26 ± 5.45 days (IQR-21–32) (Figure 7). All the patients were treated with standard COVID-19 protocol (Table 1). During hospital stays, patients were treated with standard therapy and patients in the hypercreatinemic group received more corticosteroids compared to the normocreatinemic sub-population. The clinical outcome as per creatinine concentration is presented in Table 3, which shows that the rate of intensive care unit admission, need for mechanical ventilation and in-hospital all-cause mortality were significantly higher in the hypercreatinemic group compared to the normocreatinemic sub-population. In a risk-adjusted Cox regression analysis, the hypercreatinemic group had a higher incidence of severe disease manifestation compared to the normocreatinemic sub-population (Figure 8). In the present study, an attempt was made to identify independent risk factors for all-cause, in-hospital mortality using Cox regression analysis (Table 4). From univariate analyses, old age (HR-12.83, 95% CI-2.76–15.78; *p* ≤ 0.05), elevated creatinine concentration (HR = 15.56, 95% CI = 10.56–19.67, *p* ≤ 0.001), IL-1 (9.23, 3.67–15.67, *p* ≤ 0.001), D-Dimer (HR = 4.56, 95% CI = 1.90–9.78, *p* ≤ 0.05) and Hs-Crp (HR-13.56; 95% CI 7.45–17.56, *p* ≤ 0.05) were found to be associated with all-cause in-hospital mortality. On adjusted analysis, the association of creatinine levels remained strongly predictive of all-cause, in-hospital mortality (HR-1.141 CI-1.098–1.185 *p* ≤ 0.001).

## 4. Discussion

The outbreak of COVID-19 in December 2019 created disasters in almost every country and compelled countries to undertake drastic measures ranging from complete lockdowns, identification of effective therapies, development of vaccinesto the identification of risk factors for increased susceptibility and outcome of diseases in COVID-19 patients [23]. Many independent risk factors have been identified for the grave prognosis of COVID-19, which includes old age, cardiovascular diseases, diabetes mellitus, hypertension and chronic lung diseases [24]. It is of prime importance to understand how kidney failure affects the outcomes of COVID-19 patients with concurrent kidney failure. Earlier studies have offered conflicting results because of the small sample size, incomplete follow-up, and not focusing primarily on survival considering kidney dysfunctions as a primary risk factor [25,26]. Rather, direct causality between kidney failure and COVID-19 disease severity cannot be inferred from these studies, but uremia resulting from kidney dysfunction can cause immune dysfunction, hyper-inflammation and coagulopathy [27,28,29].

SARS-CoV-2 virus triggers a cytokine storm response which results in a significant increase in levels of pro-inflammatory cytokines which cause tissue damage to renal tissue architecture. Patients with preexisting renal damage/insufficiency are more prone to the damaging/toxic effects of pro-inflammatory cytokines [13]. The tissue necrosis of renal tissue causes the release of cellular biomarkers from renal tissue which includes ALP, LDH, TNF-α, and CPK which further increases/fuels the cytokine storm response and ultimately a vicious circle is initiated, which results in the progression of renal failure in patients with preexisting renal failure [30,31]. WBC count and lymphocyte counts are usually altered in a wide spectrum of infectious and non-infectious diseasesand are used as immunohematological biomarkers for severe progression and outcome of disease [32,33]. In this direction, researchers have identified lymphopaenia as an independent biomarker for the severe progression of uremic syndrome in renal failure which increases the susceptibility/risk for various viral infections [34,35].

Earlier publications have found a significant association between kidney failure and mortality among COVID-19 patients and the susceptibility of these patients to COVID-19 [25,26,36]. Most of the available data was published from the Wuhan area of China, where focus of researchers was on the management of COVID-19 in subjects suffering from kidney derangement [37]. Furthermore, most of the earlier studies have attributed severe manifestation of COVID-19 to direct cellular damage caused by SARS-CoV-2 [38,39,40]. However, other factors associated with renal failure remain yet to be addressed. There are many limitations in these studies as these studies mostly focused on ICU patients and these studies did not use KDIGO criteria. Therefore, there is an urgent need to conduct a study in different geographical locations and take into accountthe severity of the disease and KDIGO criteria. The present study is a pioneer study that attempts to analyze the Indian cohort of hypercreatinemic patients with COVID-19 to address knowledge gapsand describe a wide spectrum of metabolic derangement in these patients.

In the present study, there were more hypercreatinemic COVID-19 males than females; this might be due to sex differences in immune-inflammatory response against COVID-19 in males than females [41]. In the present study, COVID-19 patients with hypercreatinemic were found to be in an older age group compared to the normocreatinemic sub-population. These findings are in accordance with a retrospective multicenter study, where patients with kidney dysfunction were in older age groups and were presented with severe manifestations of disease [42,43,44]. In the present study, most of the hypercreatinemic patients with COVID-19 were found to have hyper-inflammatory status and coagulopathies, as these patients were having elevated levels of LDH, ferritin, D-Dimer, procalcitonin, Hs-CRP, and IL-1 and IL-6 compared to hypercreatinemic patients with COVID-19. This is to be expected, as CKD patients are presented with underlying inflammation, immune dysregulation and altered levels of ACE-2 receptors [45]. Recently, a study by [23] found increased expression of ACE-2 receptors in patients with chronic renal failure. Taken together, increased expression of ACE-2 receptors and immune cum inflammatory dysregulation may explain the severity of COVID-19 in KD patients. In addition, some studies have identified the dual relationship between COVID-19 and KD. These studies have reported that COVID-19 increases the risk for developing kidney failure, which results in the sudden loss of kidney function and hence progresses the disease to grave outcomes, which results in the increased mortality in COVID-19 patients with preexisting abnormal kidney function [46,47]. This is further supported by significantly lower levels of proteins found in the present study, which might be attributed to proteinuria in hypercreatinemic patients. Furthermore, earlier studies have identified virus-like particles and RNA of SARS-CoV-2 in podocytes, which reflect replication of the virus in kidney tissue and thus proteinuria in COVID-19 patients [48]. Studies have found expression of ACE-2 receptors on renal tissue in addition to lungs, the gastro-intestinal tract and hepatic tissue, which causes a direct cytopathic effect owing to its viral predilection to these organ/tissues [49]. In addition, direct cytopathic effects [50] cause increased levels of CD68^+^ macrophages and C5b-9 depositions intubular epithelial cells of the kidneys which suggests hyper-activation of the immune-inflammatory mechanism which causes micro vascular damage [51]. Hyperactive activation of immune-inflammatory pathway results in multiple organ damage, which supports increased severity of disease in COVID-19 patients with KD. Furthermore, SARS-CoV-2 has been isolated from kidney tissue and urine, which indicates kidneys as potential target for SARS-CoV-2 [52]. This is further supported by a higher incidence of thromboembolic events in COVID-19 [53].

The present study reports higher mortality in hypercreatinemic patients compared to normocreatinemic patients, these findings are in accordance with [54]; they reported 12–14 times pneumonia related deaths in KD patients with COVID-19 compared to the general population. From the results of the present study, it can be postulated that kidney damage acts as an independent risk factor for the severity of the disease. To support this presumption, creatinine levels at the time of hospitalization were found to be directly associated with intensive care admissions and mechanical ventilation. Although the exact mechanism for higher mortality or the severe progression of diseases in KD patients is unknown, it is postulated that angiotensin-converting enzyme (ACE) and dipeptidyl peptidase-4 act as receptor sites for SARS-CoV-2 and studies have found enhanced expression of these receptors in failing kidneys [55,56]. Recently, autopsy of kidney samples have reveled erythrocyte aggregation and occlusion of micro-vasculature presumably caused by activated cytokine storm and compliment systems [49,57].

Multivariable analysis in the present study identified age, elevated creatinine concentration; IL-1, D-Dimer and Hs-Crp as independent risk factors for in-hospital mortality. Old age is usually associated with other co-morbidities and immune dysfunction, hence it might be an important factor for viral replication and subsequent severe manifestation of disease [41,58]. Recently, Meta-analysis involving 40 studies demonstrated KD is associated with increased mortality [29]. The present study demonstrates that higher Cr levels are associated with decreased cumulative survival rates compared to those patients with Cr levels below threshold levels.

## 5. Conclusions

In conclusion, these findings indicate that kidney dysfunction has an independent association to cause increased mortality and kidney dysfunction and may be an indicator for severe manifestations of COVID-19. Furthermore, results of the present study indicate that patients with KD need special attention with regard to COVID-19 as these categories of patients are presented with other co-morbidities and have an altered immune response to infection. In addition, these patients should take extra precautions and prevent themselves from contacting infection. Clinician should closely monitor COVID-19 patients with KD for signs of disease progression. Hence, results indicate a necessary call for the nephrology community to collect epidemiological from KD patients for better understanding of progression and outcome in realms with COVID-19.

## 6. Limitation

i.The present study, being a retrospective observational study and causal relationship between hyper creatinemic and COVID-19 outcomes, cannot be generalized. Henceforth, there is need for case controlled randomized clinical trials to further validate the role of hypercreatinemia in outcomes of COVID-19.ii.The study was conducted at a single center and hence our results suggest that there is need for a large sample size study to validate and generalize the results observed in the current study.iii.The present study excluded the patients that developed renal failure during hospitalization; we could not include these patient populations as a separate group.iv.In the present study we could not retrieve the previous information of patients with renal failure which we could not incorporate in the present manuscript.v.We could not evaluate the reason for renal failure in patients with significantly higher levels of creatinine; furthermore, we could not categorize patients into aacute renal failure group and chronic renal failure group.vi.The study was conducted from March 2020 to 15 April 2021, and during this period various variants/strains of SARS-CoV-2 were found in the COVID-19 population, so there is every possibility of a dissimilar symptomatology observed during this period.vii.In the present study, we could not measure the body weight and muscle mass of the patients and could not use these parameters for correlation studies.viii.We could not categorize patients withhypercreatinemiainto acute kidney injury, Chronic Kidney Disease, End Stage Renal Disease (ERSD) or patients with kidney transplantation.

## Figures and Tables

**Figure 1 healthcare-11-00944-f001:**
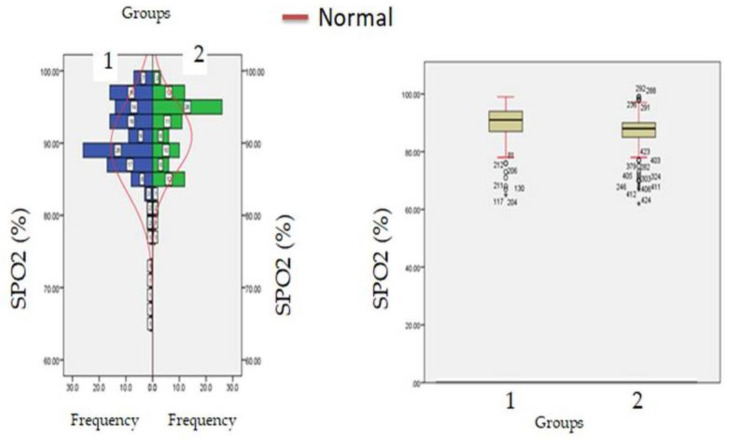
SpO_2_ levels at admission (hospitalization) in patients with normocreatinemia (Group-I) and hypercreatinemia (Group-2).

**Figure 2 healthcare-11-00944-f002:**
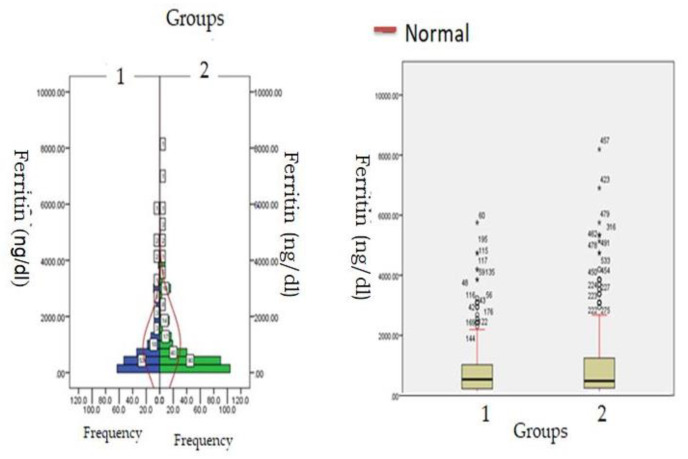
Ferritin levels at admission (hospitalization) in patients with normocreatinemia (Group-1) and hypercreatinemia (Group-2).

**Figure 3 healthcare-11-00944-f003:**
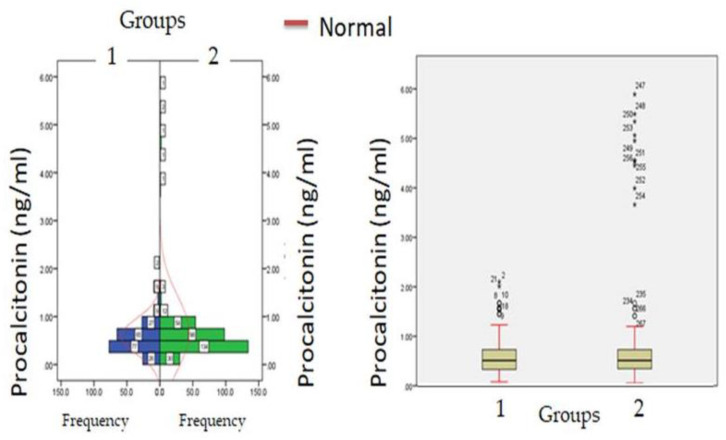
Procalcitonin levels at admission (hospitalization) in patients with normocreatinemia (Group-I) and hypercreatinemia (Group-2).

**Figure 4 healthcare-11-00944-f004:**
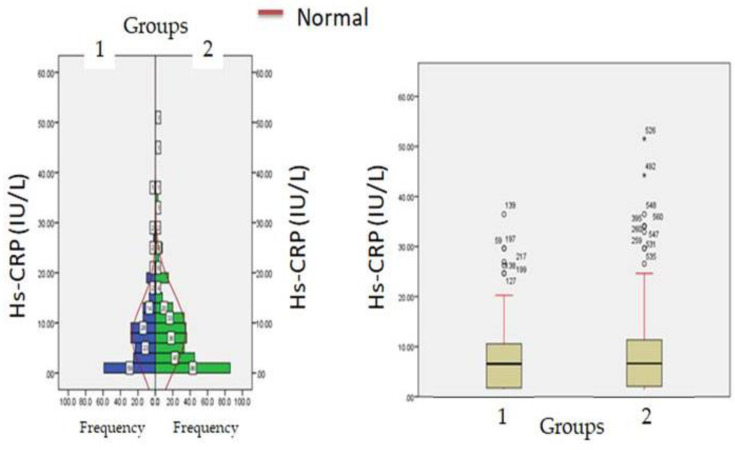
Hs-CRP levels at admission (hospitalization) in patients with normocreatinemia (Group-I) and hypercreatinemia (Group-2).

**Figure 5 healthcare-11-00944-f005:**
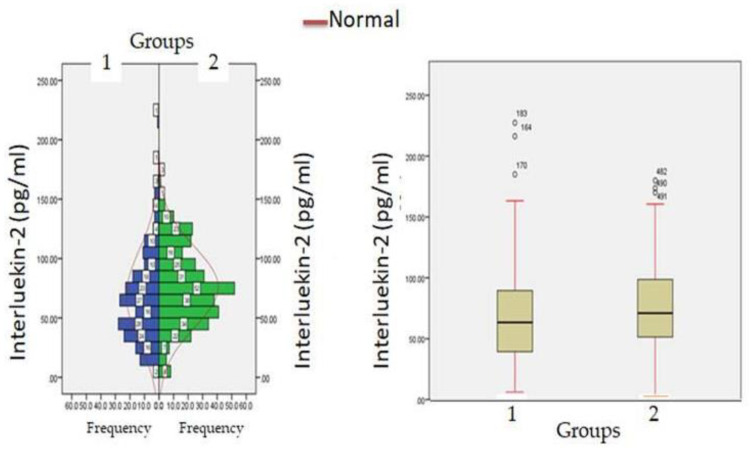
Interluekin-2 levels at admission (hospitalization) in patients with normocreatinemia (Group-I) and hypercreatinemia (Group-2).

**Figure 6 healthcare-11-00944-f006:**
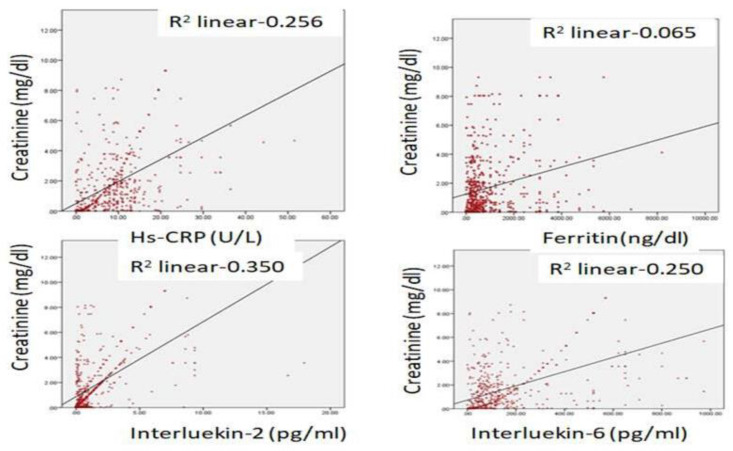
Regression analysis between admission creatinine levels and admission Hs-CRP, ferritin, IL-2 and IL-6 levels.

**Figure 7 healthcare-11-00944-f007:**
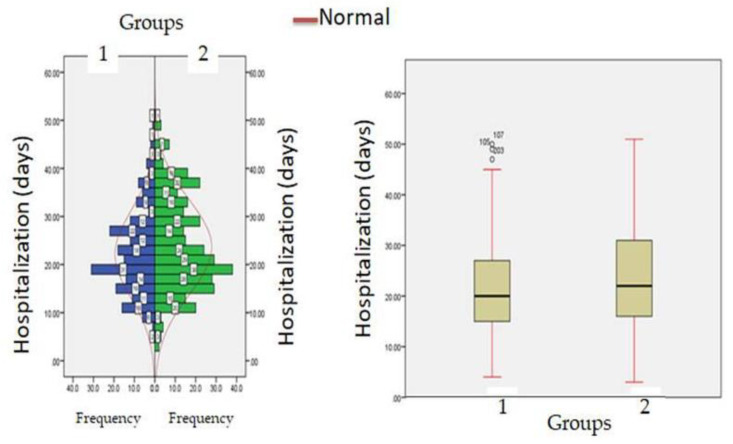
Hospitalization (days) in patients with normocreatinemia (Group-I) and hypercreatinemia (Group-2).

**Figure 8 healthcare-11-00944-f008:**
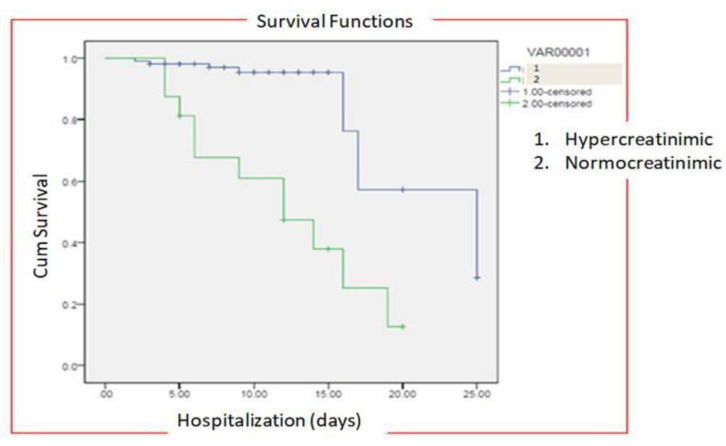
Risk-adjusted Cox regression analysis curves through 25 days for patients with COVID-19 stratified by normocreatinemia (Group-I) and hypercreatinemia (Group-2).

**Table 1 healthcare-11-00944-t001:** Comparison of laboratory indices between COVID-19 infected hypercreatinemic and normocreatinemic sub-populations.

Parameters	Group I: Normocreatinemic (*n* = 339)		Group II: Hypercreatinemic (*n* = 772)		
	95% Confidence Interval Lower Bound-Upper Bound	Mean ± SE	95% Confidence Interval Lower Bound-Upper Bound	Mean ± SE	*p*-Values
Heart Rate (beats/min)	(64.32–92.78)	87.65 ± 2.04	(79–111)	81.45 ± 8.34	0.23
Temperature (°C)	(35.40–41.84)	36.76 ± 0.39	(36.37–44.86)	38.24 ± 0.40	≤0.05
Respiratory Rate (breaths/min)	(21.12–29.67)	22.19 ± 5.83	(19.23–31.57)	29.53 ± 4.78	0.03
SPO_2_ (%)	(84.17–92.78)	89.87 ± 3.14	(85.34–96.22)	83.56 ± 5.23	≤0.01
White Blood Cell Count (10^6^/L)	(8257–9083)	8234.16 ± 512.34	(6157–9533)	6277.12 ± 871.23	≤0.05
Lymphocyte (10^6^/L)	(1134.12–1945.12)	1522.3 ± 213.3	(1256–2145)	1456.1 ± 312.1	≤0.05
Platelet Count (10^9^/L)	(112.23–207.12)	187.23 ± 24.45	(134.55–201.34)	178.51 ± 23.34	0.23
Blood Sugar (mg/dL)	(118.22–190.34)	156.45 ± 23.45	(91.34–182.11)	145.23 ± 28.90	0.11
Protein (mg/dL)	(5.45–9.34)	8.24 ± 0.76	(3.45–7.45)	6.614 ± 0.32	*p* ≤ 0.05
SGOT (IU/L)	(69.23–167.34)	78.13 ± 16.67	(79.45–123.11)	102.13 ± 14.90	≤0.001
SGPT (IU/L)	(78.23–119.45)	61.09 ± 13.08	(74.34–116.45)	92.34 ± 14.23	≤0.01
ALP (IU/L)	(106.87–189.34)	98.03 ± 18.15	(134.45–156.23)	145.34 ± 13.34	≤0.001
LDH (IU/L)	(456.23–1012.13)	452.79 ± 45.34	(858.23–972.45)	702.13 ± 34.56	≤0.001
IL-1 (pg/mL)	(0.23–2.16)	1.07 ± 0.32	(0.45–3.45)	1.87 ± 0.43	≤0.01
IL-6 (pg/mL)	(78.23–167.56)	90 ± 17.22	(102.34–456.34)	203.34 ± 25.87	≤0.01
TNF-α (pg/mL)	(6.78–87.45)	34.25 ± 21.74	(12.56–103.43)	53.56 ± 14.45	0.09
CPK (IU/L)	(451.45–1134.45)	434.45 ± 56.56	(513.14–902.23)	509.34 ± 56.35	≤0.001

SGOT: Serum glutamic oxaloacetic transaminase, SGPT: Serum glutamic pyruvic transaminase, ALP: Alkaline Phosphatase, LDH: Lactate Dehydrogenase.

**Table 2 healthcare-11-00944-t002:** Baseline characteristics of COVID-19 infected hypercreatinemic and normocreatinemic sub-populations.

Characteristics n (%)	Group I: Normocreatinemic (*n* = 339)	Group II: Hypercreatinemic (*n* = 772)	*p*-Value
Age years	48.00 ± 4.05	67.43 ± 4.36	≤0.05
Male	256 (33.16)	189 (55.75)	≤0.001
Cough	456 (59.06)	187 (51.16)	0.10
Sputum	363 (47.02)	201 (60.17)	≤0.01
Sore throat	318 (41.19)	145 (42.77)	0.13
Fever	193 (25)	194 (57.22)	≤0.01
Rhinitis	143 (18.52)	106 (31.26)	≤0.05
Insomnia	213 (11.13)	34 (10.02)	0.13
Hymoptypsis	149 (19.30)	56 (16.51)	0.89
Nausea	56 (7.25)	23 (6.78)	0.68
Diarrhoea	18 (2.33)	15 (4.42)	≤0.05
Myalgia	103 (13.34)	23 (6.78)	0.56
Fatigue	63 (8.16)	17 (5.01)	0.89
Headache	53 (6.86)	71 (20.94)	≤0.05
Azithromycin	606 (78.49)	249 (73.45)	0.65
Ivermectin	175 (52.72)	198 (58.40)	0.23
Doxycycline	507 (65.67)	244 (71.97)	0.45
Corticosteroids	134(17.33)	213 (62.83)	≤0.01
Prednisolone	125 (16.19)	123 (36.28)	0.67

**Table 3 healthcare-11-00944-t003:** Clinical outcomes of study population according to creatinine levels.

Characteristics n (%)	Group I (*n* = 339)	Group II (*n* = 772)	*p*-Value
ICU Admission (%)	224 (66.07)	269 (34.84)	≤0.001
Mechanical Ventilator Administration (%)	114 (33.62)	151 (19.55)	≤0.001
In-hospital mortality (%)	35 (10.32)	48 (6.21)	≤0.001

**Table 4 healthcare-11-00944-t004:** Cox proportional hazard analysis to identify risk factors for in-hospital mortality.

Variable	Univariable			Multivariable		
	HR	95% CI	*p* Value	HR	95% CI	*p* Value
Age (years)	12.83	2.76–15.78	≤0.05	2.72	1.09–8.67	0.047
D-Dimer (µg/mL)	4.56	1.90–9.78	≤0.05	1.05	0.56–5.23	0.782
IL-1 (pg/mL)	9.23	3.67–15.67	≤0.001	1.89	2.45–8.92	0.086
Hs-CRP (IU/L)	13.56	7.45–17.56	≤0.05	2.03	3.56–6.19	0.022
Creatinine (mg/dL)	15.56	10.56–19.67	≤0.001	5.34	4.89–8.17	≤0.001
Procalcitonin (ng/mL)	4.57	2.09–8.45	0.02	2.71	0.67–5.23	0.105
Ferritin (ng/dL)	11.90	6.45–18.78	0.04	5.09	3.12–6.34	0.214
C-Reactive protein (mg/dL)	2.09	0.89–7.90	0.008	1.05	0.34–4.17	0.137
eGFR < 60 (mL/min/1.73 m^2^)	5.89	6.89–12.64	≤0.005	2.09	0.98–7.56	≤0.05
Lactate dehydrogenase (U/L)	1.78	1.09–14.78	0.004	0.56	0.67–5.56	0.512

## Data Availability

The datasets generated during and/or analyzed during the current study are available from the corresponding author on reasonable request.

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
