# Peer review of "Clinical Characterization and Outcomes of Patients with Hypercreatinemia Affected by COVID-19"

_healthcare, 2023, doi:10.3390/healthcare11070944_

Round 1

Reviewer 1 Report

Nice point of view on Covid-19 and kidney function from India.

in this retrospective study, Elkhalifa et al. evaluated the presentation and outcomes of patients with Covid-19 and kidney injury in a large Indian hospital. The relationship between Covid-19 and acute and chronic kidney injury has been studied, but evidence from populations like the Indian one is still lacking. The authors clearly show that patients with higher levels of serum creatinine display longer hospitalizations and higher mortality; moreover, creatinine levels are an independent cause of in-hospital mortality.

I found this paper well-written, with minor spelling errors. All the results addressed the main question, and the statistics are consistent with the data shown.

Author Response

Comment 1: I found this paper well-written, with minor spelling errors.  All the results addressed the main question, and the statistics are consistent with the data shown.

Response: Thanks sir for your positive response.

Reviewer 2 Report

I read the abstract, it is well written, and it is very practical and important information for clinician to understand the negative prognostic aspect of Acute Kidney Injury "AKI" hyper-creatinemia.
I am not clear that the authors consider any other cause of elevated cr such as diabetes, HTN, older age etc.
I would suggest looking into other co morbidities that cause elevated creatinine, super imposed with COVID.

Author Response

Comment 1: I am not clear that the authors consider any other cause of elevated cr such as diabetes, HTN, older age etc. I would suggest looking into other co morbidities that cause elevated creatinine, super imposed with COVID.

Response: Significant question has been raised by reviewer; definitely these co-morbidities are having significant effect on serum creatinine levels. But in baseline characteristics of patients in both groups we excluded patients with other associated co-morbidities. In age, significant difference was observed in both groups considered in present study to nullify effect of age on clinical outcome we conducted Multivariable and Univariable Cox proportional hazard analysis. Furthermore there was no significant difference observed in co-morbidities of both the groups considered in present study so any co-morbidity having significant effect on serum Cr levels will not affect our results. 

Reviewer 3 Report

Dear Authors,

Dear authors,

COVID-19 primarily affects the respiratory tract, may manifest with systemic inflammatory response syndrome, acute kidney injury, and multiorgan dysfunction. SARS-CoV-2 has a 10–20-fold higher binding affinity for ACE2 receptors expressed in the respiratory tract, kidneys, gastrointestinal tract, and many other organs. Some studies have reported that COVID-19 patients may display kidney damage through acute kidney injury, mild proteinuria, haematuria, or slight elevation in creatinine. But there is also important to study the impact of COVID-19 on patients with pre-existing kidney impairment, including those with chronic kidney disease, kidney transplant recipients, and individuals on hemodialysis. According to other published studies, this group of patients appears to be one of the most vulnerable to COVID-19 infection. I consider this topic to be very relevant, and suitable for the purposes of this journal, and the characterization of these patients will certainly help in the management of the disease in health units, prioritizing those who need more attention.

Patients and Study Design.

In your work, you highlighted that patients with KD, are most likely to suffer from other coexisting comorbidities. The exclusion criteria eliminated from the analysis the patients with more than one comorbidity “Among these 167 (10.01 121 %) had diabetes mellitus, 145 (8.69 %) had hypertension, 21 (1.25 %) had end-stage renal failure, 101 (6.05 %) has cardiovascular dysfunctions and 123 (7.37%) were having more than one co-morbidities prior to hospitalization”. Excluding all those patients referred above and categorizing the remaining 1100 patients in two groups (hyper and normocreatinemic) seems to me a very non-natural division. Obviously, we will find a big difference in these two groups of patients in relation to the outcomes of COVID-19 and other parameters according to the presence of normal or elevated creatinine values. Here you have eliminated some of the patients who will be at greater risk because, in addition to KD, they have other co-morbidities. It seems almost impossible to dissociate KD from other health conditions. Table I shows that group II had higher glucose levels. Other blood parameters should be at this table such us cholesterol levels, blood pressure…… The medical problem that led to renal dysfunction in these patients should be presented (chronic kidney disease or kidney transplant recipients, and individuals on hemodialysis….) in a table.

Patients with previous KD should have a lot of information in their clinical history about some of the parameters that were analyzed in this study. It would be important to assess how COVID-19 has impacted these values.

In this study design you refer that “In order to differentiate patients which developed kidney function during hospitalization from those patients which were admitted in pre-existing kidney dysfunction we excluded patients which developed kidney dysfunction during hospitalization.”

I think that understanding the causes of why some patients with COVID-19 develop renal complications is relevant to be able to identify patients who are at risk after infection and this sample is appropriate to analyze these findings as well.

You must clarify the discussion around figure 6 (Regression analysis between admission creatinine levels and admission Hs-CRP, ferritin, IL-2, and IL-6 levels).

Discussion

Considering that systematic sampling was done between March 2020 to 15th April 2021, you should discuss or mention one limitation of the study in the sense that different variants of SARS-CoV-2 seem to lead to dissimilar symptomatology. Could the infection of patients with the omicron variant, for example, lead to these same conclusions?

Specific Comments

Please correct or confirm:

·        “KD was defined as per (based on) serum creatinine criteria. Significant increase in serum creatinine by 1.5 times higher from the baseline values established in our referral ?? laboratory.” …. According to the Kidney Disease Improving Global Outcomes (KDIGO) criteria, AKI can be defined as an increase in creatinine to at least 1.5 times baseline creatinine (data from our reference laboratory?) that is presumed to have occurred within 7 days.

·        _(space) “Clinically disease affects the respiratory system and is characterized by acute respiratory distress syndrome, but () in addition to pulmonary …” Throughout the text you have several words that need a space between them.

·        “ Declaration of Helsinki and oral consent? (Informed consent?)

·        The severity of COVID-19 disease was defined according to the guidelines of the World Health Organization (WHO) classification for COVID-19 and the CURB-65 scale,

·        The scale of the graphics is too small, please consider adding a more detailed legend. Tables also need a detailed legend. The graphic scale of boxplots can be changed to better visualize the differences between groups and it was important adding the result of statistical analysis (differences between groups).

·        “Table 1. Comparison of laboratory indices between COVID-19 infected hyper-creatinemic and normo-creatini(e)mic sub-population.” Please use normocreatinemic and hypercreatinemic and correct the figure legends too. 

Kind Regards

Author Response

Comment 1:Here  you  have  eliminated  some  of the  patients  who  will  be  at  greater  risk  because,  in  addition  toKD,  they  have  other  co-morbidities.  It seems almost impossible to dissociate KD from other health conditions.

Response: Definitely it seems impossible to dissociate KD from other health conditions but in present study inclusion criteria we followed was “no history of co-morbidities prior to hospitalization” and we attempted to exclude those patients having known history of co-morbidities. Patients were divided in two groups only on basis of levels of Cr above threshold levels. 

Comment 2: Table I shows that group II had higher glucose levels.  Other  blood  parameters should  be  at  this  table  such  us  cholesterol  levels,  blood pressure.

Response: Although, blood sugar levels are higher in group II, but there is no significant difference between two groups. Other parameters are included in revised manuscript.

Comment 3:The  medical  problem  that  led  to  renal  dysfunction in  these patients should be presented (chronic kidney disease or kidney  transplant  recipients,  and  individuals  on  hemodialysis) in a table.

Response: we could not categorize patients based on medical problem  that  led  to  renal  dysfunction and same has been included in limitation of the study.

Comment 4:Patients  with  previous  KD  should  have  a  lot  of  information  in their  clinical  history  about  some  of  the  parameters  that  were analyzed  in  this  study .  It would be important to assess how COVID-19 has impacted these values.

Response:Sir, sorry we could not retrieve that information, although that information was used during hospitalization to examine clinical improvement of patients. Same point has been included in limitations of the study.

Comment 5:In  this  study  design  you  refer  that  “In  order  to  differentiate patients  which  developed  kidney  function  during  hospitalization from  those  patients  which  were  admitted  in  pre-existing  kidney dysfunction  we  excluded  patients  which  developed  kidney dysfunction during hospitalization.”I  think  that  understanding  the  causes  of  why  some  patients  with COVID-19  develop  renal  complications  is  relevant  to  be  able  to identify  patients who are at risk  after infection  and this  sample is appropriate to analyze these findings as well.

Response: we agree that those patients should have been included as separate group, other manuscript for that is under process and we will communicate that in near future.

Comment 6:You  must  clarify  the  discussion  around  figure  6  (Regression analysis between  admission creatinine  levels  and admission (HsCRP , ferritin, IL-2, and IL-6 levels).

Response: Discussion for figure 6 has been included in result section of revised manuscript.

Comment 7:Considering  that  systematic  sampling  was  done  between  March 2020  to  15   April  2021,  you  should  discuss  or  mention  one limitation  of  the  study  in  the  sense  that  different  variants  of SARS-CoV-2  seem  to  lead  to  dissimilar  symptomatology .  Could the infection of patients with the omicron variant, for example, lead to these same conclusions?

Response: Same has been included in limitations of study.

Reviewer 4 Report

This clinical study focuses on observing the outcomes of patients with hypercreatinemia affected by COVID-19. Some data was interesting, but some were very confusing. Therefore, several concerns have to be satisfactorily addressed.

1. Since the authors investigated the outcomes of hypercreatinemia patients, a fundamental question is the body weight and muscle content have to be measured. The correlations need to be analyzed.

2. It is surprising to see WBC and lymphocyte counts were reduced while ALP, LDH, IL-1, IL-6, TNF-a, and CPK were increased in hyper-creatinimic patients, if compared to normo-creatinemae. The authors need to further clarify and discuss it.

3. How to quantify the major symptoms shown in Table 2? A special form?

4. Figure 6, regression analysis should involve serum creatinine levels.

5. Minor, in Figures 1-5 & 7-8, the authors named Group-I as normo-creatinemae and Group 2 as hypercreaniemae. It seems not the case. It is not consistent with the text and presented tables. Please check. Line 154, the authors stated that biochemical parameters (Table 2) which include…., should be in Table 1.

6. Minor, all box plots are very confusing and easily cause potential misleading. The authors need to improve them.

Author Response

Comment 1: Since  the  authors  investigated  the  outcomes  of hypercreatinemia  patients,  a  fundamental  question  is  the  body weight  and  muscle  content  have  to  be  measured.  The correlations need to be analyzed.

Response:  It would have been an interesting finding if we could have measured body weight and muscle content of the patients. But unfortunately due to epidemic crisis we could not record body weight and muscle content. Same has been included in limitations of the study of present manuscript.

Comment 2:It  is  surprising  to  see  WBC  and  lymphocyte  counts  were reduced  while  ALP ,  LDH,  IL-1,  IL-6,  TNF-a,  and  CPK  were increased  in  hypercreatinimic  patients,  if  compared  to  normocreatinemae. The authors need to further clarify and discuss it.

Response:Response has been added in revised manuscript as high lightened with yellow color

Comment 3: How to quantify the major symptoms shown in Table 2?  A special form?

Response: these symptoms were recorded in patient record sheet and later entered in electronic database.

Comment 4:Figure 6, regression analysis should involve serum creatinine levels.

Response: it already there on Y axis.

Comment 5: Minor, in Figures 1-5 & 7-8, the authors named Group-I as normo-creatinemae and Group 2 as hypercreaniemae.  It seems not the case.  It is not consistent with the text and presented tables.  Please check.  Line  154,  the  authors  stated  that biochemical  parameters  (T able  2)  which include….,  should  be  in T able 1.

Response: Corrected in revised manuscript.

Comment 6: Minor, all box plots are very confusing and easily causepotential misleading. The authors need to improve them.

Response: All figures have been improved.

Reviewer 5 Report

Thanks for giving me the opportunity to review this hard work. Unfortunately there are a lot of corrections that have to be addressed here. Some of them are mentioned below:

-What is hypercreatinemia? its either acute kidney injury or CKD or ERSD or may a patient with a kidney transplant with any insult or may be a kidney donor? They are all very different patients and researchers have not even commented that what class are they seeing and analyzing.

-There is huge bank of data showing outcomes and courses in all these categories and none of which has been mentioned in literature review: few examples :DOI: 10.1053/j.ajkd.2020.09.003 ,

https://doi.org/10.1053/j.ajkd.2021.11.004

https://doi.org/10.1111/ajt.16280

So I don't see any extra information coming from what we are doing here.  Researchers have to describe what is the additional need to do this study ? is there a different population? 

-Study design materials and methods: I am not able to clearly understand from any table that who is group one and who is group 2 ? There should be a clear flow chart or description in methods that how many pateints are in each group. Tables and figures in scientific writing should stand as independent legends.

-There are huge flaws in the rationale of the study. What was the normal baseline creatine o these people and did they have a CKD, transplant, ESRD, or were the donors, and who did the worst? Comparing hypercreatininemia without even knowing the baseline creatinine is not a good scientific way.

Lets say patient A had a baseline cr < 1 and now has cr elevated to 5 and patient B was already on dialysis with a baseline of 5 and patient C was a kidney transplant;ant patient with creatine of 2 on immunosuppression , can they be taken as the same group

Author Response

Comment 1:What is hypercreatinemia? It is either acute kidney injury or CKD or ERSD or may a patient with a kidney transplant with any insult or may be a kidney donor? They are all very different patients and researchers have not even commented that what class are they seeing and analyzing.

Response:Animportant query has been raised by reviewer, asaim of the present study was to evaluate role of hypercreaniemae as risk factor for severe outcome of COVID-19, so we included all patients having significantly increased levels of hypercreaniemae. In addition to this we have included the comment raised by reviewer as limitation of study in “LIMITATION” section.

Comment 2:There is huge bank of data showing outcomes and courses in all these categories and none of which has been mentioned in literature review: few examples: DOI: 10.1053/j.ajkd.2020.09.003, https://doi.org/10.1053/j.ajkd.2021.1 1.004 https://doi.org/10.1 1 1 1/ajt.16280 so I don't see any extra information coming from what we are doing here. Researchers have to describe what is the additional need to do this study? Is there a different population?

Response: We have mentioned these publications in revised manuscript. The population considered in present study is Asian population while above mentioned manuscripts have conducted study in American populations.

Comment 3:Study design materials and methods: I am not able to clearly understand from any table that who is group one and who is group 2? There should be a clear flow chart or description in methods that how many patients are in each group. Tables and figures in scientific writing should stand as independent legends.

Response Corrected in revised manuscript and flow chart is provided as supplementary figure.

Comment 4: There are huge flaws in the rationale of the study. What was the normal baseline creatinine of these people and did they have a CKD, transplant, ESRD, or were the donors, and who did the worst? Comparing hypercreatininemia without even knowing the baseline creatinine is not a good scientific way.

Response: It would have been an interesting finding if we could have categorized patients into CKD, transplant, ESRD, or were the donors. We have included same in limitations of study portion.

Comment 5:Let’s say patient Ahad a baseline cr< 1 and now has cr elevated to 5 and patient B was already on dialysis with a baseline of 5 and patient C was a kidney transplant and patient with creatine of 2 on immunosuppression, can they be taken as the same group.

Response: we have included statement in limitations of study portion.

Reviewer 6 Report

This study provides important insights into the impact of elevated creatinine levels on the severity of COVID-19 and the risk of mortality. The findings suggest that patients with hyper creatinemic are more likely to experience severe manifestations of COVID-19. The association of creatinine levels with all-cause in-hospital mortality remained strongly predictive on adjusted analysis. The study's results emphasize the importance of ameliorating kidney function in patients with COVID-19, which could help achieve creatinemic targets and potentially improve outcomes.

There are some limitations that should be considered:

1.      The study is a retrospective observational study, which may limit the ability to establish causal relationships between hyper creatinemic and COVID-19 outcomes.

2.      The study was conducted in a single hospital in Srinagar, India, which may limit the generalizability of the findings to other populations and settings.

3.      The measurement of creatinine levels: The study used creatinine levels at admission to define hyper creatinemic, which may not accurately reflect the true baseline creatinine level in some patients.

4.      The study did not account for all potential confounding factors that may affect the association between hyper creatinemic and COVID-19 outcomes.

5.      The study did not provide information on the specific treatment protocols used for patients with hyper creatinemic, which may have influenced the study's findings.

Overall, while the study provides important insights into the association between hyper creatinemic and COVID-19 outcomes.

Author Response

Comment 1:The study is a retrospective observational study, which may limit the ability to establish causal relationships between hyper creatinemic and COVID-19 outcomes.

Response:An interestingcomment has been raised byreviewer;statement has been included in Limitations of the study portion. 

Comment 2:The study was conducted in a single hospital in Srinagar, India, which may limit the generalizability of the findings to other populations and settings.

Response:An interestingcomment has been raised byreviewer;statement has been included in Limitations of the study portion.  

        Comment 3:The measurement of creatinine levels: The study used creatinine levels at admission to define hyper creatinemic, which may not accurately reflect the true baseline creatinine level in some patients.

        Response: The baseline was established in hospital in healthy patient group, based on baseline values and guidelines of Kidney disease guidelines as defined by Improving Global Outcomes (KDIGO) CKD Work Group (2012) classification (KDIGO, 2012). KD was defined as per serum creatinine criteria and was defined as significant increase in serum creatinine by 1.5 times higher from the baseline values established in Chest Disease Laboratory, Srinagar Jammu and Kashmir.

Comment 4:The study did not account for all potential confounding factors that may affect the association between hyper creatinemic and COVID-19 outcomes.

Response:Multivariable analysis revealed age, elevated creatinine concentration; IL-1, D-Dimer and Hs-Crp were independent risk factors for in-hospital mortality. On adjusted analysis, the association of creatinine levels remained strongly predictive of all cause in-hospital mortality (HR-5.34; CI-4.89-8.17; p ≤ 0.001).Furthermore, we could not find any significant difference in other co-morbidities which might have effect on levels of creatinine.In addition to this, in present study inclusion criteria we followed was “no history of co-morbidities prior to hospitalization” and we attempted to exclude those patients having known history of co-morbidities. Patients were divided in two groups only on basis of levels of Cr above threshold levels.

Comment 5: The study did not provide information on the specific treatment protocols used for patients with hyper creatinemic, which may have influenced the study's findings.

Response: All the patients were treated with standard COVID-19 protocol (Table 1). During hospital stay patients were treated with standard therapy and patients in hypercreatinemic group received more corticosteroids compared to normocreatinemic sub-population.

Round 2

Reviewer 4 Report

Thanks for the authors' response. I have no additional comments.

Author Response

(The authors gave the same response as above.)
